# γ-ray and ν Searches for Dark-Matter Subhalos in the Milky Way with a Baryonic Potential

**Moritz Hütten** [1,*] , **Martin Stref** [2,*] , **Céline Combet** [3] **and Julien Lavalle** [2] **and David Maurin** [3]

[1] Max-Planck-Institut für Physik, Föhringer Ring 6, D-80805 München, Germany
[2] Laboratoire Univers & Particules de Montpellier (LUPM), CNRS & Université de Montpellier (UMR-5299), Place Eugène Bataillon, F-34095 Montpellier CEDEX 05, France; lavalle@in2p3.fr
[3] Laboratoire de Physique Subatomique et de Cosmologie, Université Grenoble-Alpes, CNRS/IN2P3, 53 Avenue des Martyrs, 38026 Grenoble, France; celine.combet@lpsc.in2p3.fr (C.C.); dmaurin@lpsc.in2p3.fr (D.M.)
[*] Correspondence: mhuetten@mpp.mpg.de (M.H.); martin.stref@umontpellier.fr (M.S.)

**Abstract:** The distribution of dark-matter (DM) subhalos in our galaxy remains disputed, leading to varying γ-ray and ν flux predictions from their annihilation or decay. In this work, we study how, in the inner galaxy, subhalo tidal disruption from the galactic baryonic potential impacts these signals. Based on state-of-the art modeling of this effect from numerical simulations and semi-analytical results, updated subhalo spatial distributions are derived and included in the CLUMPY code. The latter is used to produce a thousand realizations of the γ-ray and ν sky. Compared to predictions based on DM only, we conclude a decrease of the flux of the brightest subhalo by a factor of 2 to 7 for annihilating DM and no impact on decaying DM: the discovery prospects or limits subhalos can set on DM candidates are affected by the same factor. This study also provides probability density functions for the distance, mass, and angular distribution of the brightest subhalo, among which the mass may hint at its nature: it is most likely a dwarf spheroidal galaxy in the case of strong tidal effects from the baryonic potential, whereas it is lighter and possibly a dark halo for DM only or less pronounced tidal effects.

**Keywords:** dark matter; galactic subhalos; semi-analytic modeling; gamma-rays and neutrinos

---

## 1. Introduction

In this contribution to *The Role of Halo Substructure in Gamma-Ray Dark-Matter Searches*, we revisit a previous study on the detectability of galactic dark clumps in γ-rays [1]. The latter relied on the best knowledge we had a few years ago of the properties of dark-matter (DM) clumps in the Milky Way. These properties (e.g., the mass and spatial distributions of galactic subhalos) were inferred from numerical simulations with a typical mass resolution of a few $10^5$ M$_\odot$, and extrapolated down ten orders of magnitude to the model-dependent minimal masses of subhalos [2,3]. Functional parametrizations were incorporated in the CLUMPY code [4–6] to generate γ-ray skymaps, accounting for the whole population of subhalos. For each combination of subhalo properties we explored, hundreds of skymap realizations were drawn, allowing us to calculate the average properties of the brightest clump. In the context of the future CTA γ-ray observatory [7] and its foreseen extragalactic survey, we concluded that the limits on DM set from this brightest clump should be "competitive and complementary to those based on long observation of a single bright dwarf spheroidal galaxy".

In the recent years, numerical simulations [8–10] and semi-analytical studies [11–15] have investigated the impact of the baryonic components of disk galaxies on their subhalo population

by tidal stripping and disruption. These works reached the generic conclusion of a strong depletion of subhalos in the disk regions (i.e., also the Solar neighborhood), though with different quantitative estimates. Such a difference is expected due to the diversity of assumptions and methods used by different groups. In any case, this immediately questions the conclusion reached in [1], where the brightest subhalo was found, on average, at ∼10 kpc from the Galactic center and at similar distance from Earth. Experimental limits on DM from galactic subhalos, derived from *Fermi*-LAT [16–24] or expected from prolonged operation [25], from HAWC [26,27], or future instruments such as GAMMA 400 [28] and CTA [1,29], should also be impacted by this result.

The paper is organized as follows: Section 2 presents the overall methodology and recalls how all relevant subhalos are efficiently accounted for with CLUMPY. Section 3 lists the subhalo critical parameters, highlighting the very different spatial distributions considered in this analysis. Section 4 presents updated statistics of the subhalo population and provides probability density functions (PDFs) of the brightest subhalo's properties (distance to the observer, mass, brightness, etc.). The analysis is performed for both annihilating DM [30–32] via the so-called *J*-factors, or decaying DM [33,34] (*D*-factors). We also show one realization of a subhalo skymap for all configurations considered. We conclude and briefly comment on the consequences for DM indirect detection limits in Section 5.

## 2. Important Quantities and Methodology

### 2.1. γ-ray and ν Fluxes from Dark Matter

The γ-ray or ν flux from annihilating/decaying DM particles, at energy $E$, along the line-of-sight (l.o.s.) in the direction $(\psi, \theta)$, and integrated over the solid angle $\Delta\Omega = 2\pi \left(1 - \cos \alpha_{\text{int}}\right)$, is given by

$$\frac{\mathrm{d}\Phi_{\gamma,\nu}}{\mathrm{d}E}\left(E, \psi, \theta, \Delta\Omega\right)^{\text{Annihil.}}_{\text{Decay}} = \underbrace{\frac{1}{4\pi} \sum_f \frac{\mathrm{d}N^f_{\gamma,\nu}}{\mathrm{d}E} B_f \times \begin{Bmatrix} \dfrac{\langle \sigma v \rangle}{m^2_{\text{DM}} \delta} \\[2mm] \dfrac{1}{\tau_{\text{DM}} \, m_{\text{DM}}} \end{Bmatrix}}_{\text{Particle physics term: } \frac{\mathrm{d}\Phi^{PP}_{\gamma,\nu}}{\mathrm{d}E}(E)} \times \underbrace{\int_0^{\Delta\Omega} \int_{\text{l.o.s.}} \mathrm{d}l \, \mathrm{d}\Omega \times \begin{Bmatrix} \rho^2(\vec{r}) \\[2mm] \rho(\vec{r}) \end{Bmatrix}}_{\text{Astrophysics term: } J\text{- or } D\text{-factor}}, \quad (1)$$

where we made the distinction between the cases of DM self-annihilation (top) and decay (bottom). In both cases, $m_\chi$ is the mass of the DM particle, $\mathrm{d}N^f_\gamma/\mathrm{d}E$ and $B_f$ correspond to the spectrum per interaction and branching ratio of annihilation or decay channel $f$, and $l$ is the distance from the observer. In case of annihilation, $\langle \sigma v \rangle$ is the velocity-averaged cross-section[1], while $\tau_{\text{DM}}$ is the DM particle lifetime in the decay scenario. Finally, $\rho(\vec{r})$ is the overall Galactic DM density distribution. The latter can be cast formally as the sum of a smooth distribution $\rho_{\text{sm}}$ of unclustered DM particles, and a collection of $i = 1 \ldots N_{\text{subs}}$ subhalos, each with a density $\rho_i(\vec{r})$. The astrophysical term for annihilating DM[2] then reads [4,5,35],

$$\frac{\mathrm{d}J_{\text{tot}}}{\mathrm{d}\Omega} = \int_{\text{l.o.s.}} \left(\rho_{\text{sm}} + \sum_i^{N_{\text{subs}}} \rho_i\right)^2 \mathrm{d}l = \underbrace{\iint \rho^2_{\text{sm}} \, \mathrm{d}l}_{\mathrm{d}J_{\text{sm}}/\mathrm{d}\Omega} + \underbrace{\iint \left(\sum_i^{N_{\text{subs}}} \rho_i\right)^2 \mathrm{d}l}_{\mathrm{d}J_{\text{subs}}/\mathrm{d}\Omega} + \underbrace{2\iint \rho_{\text{sm}} \sum_i^{N_{\text{subs}}} \rho_i \, \mathrm{d}l}_{\mathrm{d}J_{\text{cross-prod}}/\mathrm{d}\Omega}. \quad (2)$$

The above formula corresponds to a single realization of the underlying distribution of subhalos in the galaxy. The statistical properties of this distribution can be partly obtained from the formalism of hierarchical structure formation (e.g., [36]) or extracted from numerical simulations, as discussed below.

---

[1]  We assume here the DM particle to be a Majorana particle, so that $\delta = 2$ (for a Dirac, $\delta = 4$).
[2]  For conciseness, we present in Equations (2) and (3) formulae for annihilating DM only. Analogous formulae without a cross-product term and linear in the DM density can also be written for the *D*-factor of decaying DM.

*2.2. Generating Skymaps with* `CLUMPY` *v3.0*

For all our calculations, we rely on the public `CLUMPY` code described in [4–6]. It is a flexible code that efficiently emulates the end-product of numerical simulations in terms of $\gamma$-ray and neutrino signals for DM annihilation or decay. It allows easy exploration of how results are affected when changing the properties of the DM halos. `CLUMPY` v2.0 [5] was used for this purpose, to estimate the sensitivity of the CTA [1] and HAWC [27] $\gamma$-ray telescopes to galactic DM subhalos. Aside from galactic subhalo studies, `CLUMPY` has also been used by several teams to model DM annihilation or decay in $\gamma$-rays or $\nu$ in many targets: dwarf spheroidal galaxies [37–46], the galactic halo [47–49], the Smith HI cloud [50], nearby galaxies [51], galaxy clusters [52–54], and also for the extragalactic diffuse emission [55].

The present analysis is performed with `CLUMPY` v3.0 [6].[3] For completeness, we recap below the main steps of the `CLUMPY` calculation used for this work:

- `CLUMPY` *and the particle physics term:* Equation (1) shows that the particle physics term and the astrophysical terms are decoupled.[4] As the flux depends on the specific DM candidate chosen, we provide results in terms of $J$- and $D$-factors only; `CLUMPY` can easily be used to transform those into $\gamma$-ray or $\nu$ fluxes for any user-defined DM candidate (see `CLUMPY`'s online documentation[5]).
- `CLUMPY` *and the astrophysics term:* to calculate skymaps of $dJ/d\Omega$, one should rely in principle on Equation (2). However, this is impractical in terms of computing time, as $\sim 10^{14}$ subhalos are expected in a Milky Way-sized DM halo. This problem can be overcome by formally separating Equation (2) in an average and "resolved" component,

$$\frac{dJ_{\text{tot}}}{d\Omega} = \frac{dJ_{\text{sm}}}{d\Omega} + \left\langle \frac{dJ_{\text{subs}}}{d\Omega} \right\rangle + \left\langle \frac{dJ_{\text{cross-prod}}}{d\Omega} \right\rangle + \sum_{k=1}^{N_{\text{subs}}^{\text{drawn}}} \frac{dJ_{\text{drawn}}^k}{d\Omega}. \tag{3}$$

With this ansatz, only a limited number $N_{\text{subs}}^{\text{drawn}}$ of subhalos need to have their $J$-factor profiles calculated individually, while an average description is sought for the remaining "unresolved" DM. The criterion to discriminate between resolved and unresolved components often relies on a simple subhalo mass threshold, e.g., as done in works directly relying on numerical simulations [57] or their subhalo catalogs [58]. `CLUMPY` has been developed to treat this problem in a more efficient way, acknowledging the fact that rather light, but close-by subhalos may show $J$-factors comparable to heavier, more distant objects. The `CLUMPY` approach relies on the notion that the overall DM signal fluctuates around an average description, $\langle J_{\text{tot}} \rangle \pm \sigma_{J_{\text{subs}}}$, and we refer to [4] for a detailed description of our criterion to accordingly discriminate between unresolved and resolved halos. For the purpose of this work (and also the previous [1]), this approach allows us to preselect halos likely to shine bright at Earth and to consider all decades down to the smallest subhalo masses in the calculation.

## 3. Modeling the Galactic Subhalo Distribution

In this study, we focus on the impact of tidal disruption of subhalos in interaction with the baryonic components of the Milky Way, and compare four parametric models of the resulting galactic subhalo

---

[3]    When releasing `CLUMPY` v3.0, we corrected a misprint that was present in v2.0, related to our implementation of the virial overdensity from [56]. This issue was responsible for obtaining in [1] about a factor 3 more subhalos than expected per flux decade (see full details in the `CLUMPY` documentation). We recall that in [1], we found that galactic variance is responsible for a factor $\lesssim 10$ uncertainty on the value of the brightest subhalo, and that other subhalo properties were responsible for another factor $\sim 6$. Given these very large uncertainties, the conclusions on DM limits set from dark clumps with `CLUMPY` v2.0 are not qualitatively changed, but we urge users to rely on `CLUMPY` v3.0 for future studies.

[4]    Strictly speaking, this factorization holds true only for DM candidates for which $\langle \sigma v \rangle$ is independent of the velocity and consideration of small redshift cells, $\Delta z/z \ll 1$.

[5]    https://lpsc.in2p3.fr/clumpy.

abundance and their $J/D$-factors. We consider three quantities to be most sensitive to the $J/D$-factor distribution: (i) the spatial PDF of subhalos in the Milky Way, $\mathrm{d}\mathcal{P}/\mathrm{d}V$, (ii) the mass-concentration[6] relationship, $c(m, r)$, and (iii) the calibration of the total number of subhalos in the Milky Way, $N_\mathrm{calib}$. The latter number is determined from numerical simulations, in a range where subhalos are resolved. Here, $N_\mathrm{calib}$ is defined for the mass range $10^8$–$10^{10}$ $\mathrm{M}_\odot$, and it typically falls in the range 100–300.[7]

For modeling the subhalo distribution with these parameters, we start from an "unevolved" distribution, where we assume the position and mass of a subhalo to be uncorrelated,

$$\frac{\mathrm{d}^3\mathcal{P}}{\mathrm{d}V\mathrm{d}m\,\mathrm{d}c} = \frac{\mathrm{d}\mathcal{P}}{\mathrm{d}V}(\vec{r}) \times \frac{\mathrm{d}\mathcal{P}}{\mathrm{d}m}(m) \times \frac{\mathrm{d}\mathcal{P}}{\mathrm{d}c}(c, m). \tag{4}$$

Here, $\mathrm{d}\mathcal{P}/\mathrm{d}m$ and $\mathrm{d}\mathcal{P}/\mathrm{d}c$ describe the PDFs for a subhalo to have a given mass and a given concentration $c$. In reality, the factorization in Equation (4) may break down when subhalos gravitationally interact with the DM and baryonic potentials of their host halo [15,62], entangling their mass and positional distributions. We will consider this effect by "evolving" the distribution of Equation (4) in the model presented in Section 3.2.3.

## 3.1. Fixed Subhalo-Related Quantities

This work focuses on the impact of a baryonic disk potential on the subhalo population, mainly through the spatial PDF $\mathrm{d}\mathcal{P}/\mathrm{d}V$ (see Section 3.2). We keep several other subhalo-related quantities fixed to isolate this effect. For details on these parameters and how they affect the subhalo emission, we refer to our earlier work in [1] and only provide a brief summary below:

- *Index $\alpha_m$ of the power-law subhalo mass PDF $\mathrm{d}\mathcal{P}/\mathrm{d}m \propto m^{-\alpha_m}$ and subhalo mass range:* We choose $\alpha_m = 1.9$, $m_\mathrm{min} = 10^{-6}\,\mathrm{M}_\odot$, and $m_\mathrm{max} = 1.3 \times 10^{10}\,\mathrm{M}_\odot$. The maximum clump mass for all models is set to $10^{-2} \times M_{200}$ of the NFW halo from Section 3.2.3. This is motivated by the fact that we do not consider the possibility of any subhalos heavier than the Magellanic clouds, the heaviest satellites of our galaxy. The minimal clump mass and $\alpha_m$ mostly affect the diffuse emission boost from unresolved halos. For a fixed normalization $N_\mathrm{calib}$, a steeper mass function ($\alpha_m = 2.0$) decreases the number of bright halos ($J \gtrsim 10^{20}\,\mathrm{GeV}^2\,\mathrm{cm}^{-3}$) by not more than ~30%.
- *Width of $\mathrm{d}\mathcal{P}/\mathrm{d}c$:* We set $\sigma_c = 0.14$ [63]. Using a larger scatter $\sigma_c = 0.24$ [64] increases only by a few percent the number of subhalos per flux decade. Reducing the scatter or no scatter has the opposite effect.
- *Subhalo density profile:* We model all subhalos with a spherically symmetric NFW profile [65]. Using an Einasto profile [65,66] instead amounts to a global increase ~2 of the number of subhalos per flux decade within the considered integration regions $\Delta\Omega$. Please note that micro-halos with $m \ll \mathrm{M}_\odot$ may show steeper inner slopes [67–69]; however, we have found that these micro-halos do not provide new bright, resolved subhalo candidates [1].
- *Level of sub-substructures:* We do not consider an emission boost from substructure within subhalos. Such a boost from additional levels of substructure[8] increases the number of subhalos per flux decade, with the largest increase of almost a factor 2 for the largest luminosities. Sub-substructures actually increase the signal in the outskirts of halos (see Figure 4 of [1]), the impact of which depends on the instrument angular resolution or containment angle used in the analysis. For instance, in [1], no impact was found for dark clumps within the angular resolution of CTA.

---

[6]   The concentration is defined to be $c = r_\Delta/r_{-2}$, with $r_\Delta$, taken to be the subhalo boundary, is the radius at which the mean subhalo density is $\Delta$ times the critical density (see, e.g., [6]), and $r_{-2}$ is the position in the subhalo for which the slope of the density is $-2$. We use $\Delta = 200$ in this work.

[7]   This range was recently shown to be in agreement with the observed number of dwarf spheroidal galaxies SDSS corrected by the detection efficiency [59], alleviating the tension caused by the so-called missing satellite problem in CDM scenarios [60,61]. Given the minimal mass of subhalos, $m_\mathrm{min}$, $N_\mathrm{calib}$ can be used to calculate the mass fraction, $f_\mathrm{DM}$, of DM in subhalos.

[8]   As shown in [5] (see their Figure 1), only the first level of substructure significantly boosts the halo luminosity, the next levels bringing a few extra percent at most.

Please note that our choice for these constant parameters will lead to a rather conservative number of detectable subhalos and *J*-factor of the brightest 'resolved' subhalo—hence conservative limits for DM indirect detection—compared to other choices. Pushing all the parameters to get the most optimistic case would lead to a factor ∼2–3 increase of the *J*-factor of the brightest subhalo [1].

### 3.2. The Spatial Distribution d$\mathcal{P}$/d$V$ of Subhalos

We consider four configurations of d$\mathcal{P}$/d$V$ in this work, which are described below and summarized in Table 1. The first configuration (model #1) is close to one of our 2016 study [1], i.e., based on results from DM-only simulations. It is used as a reference to which the other configurations describing interaction with the baryonic disk are compared to. We consider a spherically symmetric Galactic DM halo and correspondingly, also d$\mathcal{P}$/d$V$ distribution. The maximum distance of any subhalo from the Galactic center is set to $R_{200} = 231.7$ kpc for all configurations, inspired by the NFW halo from Section 3.2.3. We show later that the brightest subhalo is found only with negligible probability at larger Galactocentric radii for all models. Please note that despite the common value for $R_{200}$, the total Galactic DM profile is different from one configuration to another. We however focus here on the clumpy part of the halo only and do not consider the smoothly distributed DM.[9] While this article is dedicated to the derivation and comparison of statistical properties of the subhalo population brightness for these different models, we still emphasize that when going to firm predictions and limits, the overall DM profile should matter. Indeed, the Milky Way is a strongly constrained system [70,71], which must be taken into account when extrapolating simulation results. This aspect goes beyond the scope of this work though, but is worth mentioning as the Gaia mission is currently boosting our handle on Milky Way dynamics [72,73].

**Table 1.** Subhalo parameters for the models investigated in this study, with model #1 based on results of DM-only numerical simulations, while models #2 to #4 are different implementations of DM subhalos post-processed in the Milky Way halo and baryonic disk potential. For models #2, #3, and #4, we also show the number of surviving subhalos with tidal masses between $10^8$ M$_\odot$ and $10^{10}$ M$_\odot$. See Section 3 for details and parameters common to all subhalo configurations.

| | Model #1 | Model #2 | Model #3 | Model #4 |
|---|---|---|---|---|
| $\dfrac{\mathrm{d}\mathcal{P}}{\mathrm{d}V}$ | Aquarius [74] Einasto $\alpha_E = 0.68$ $r_{-2} = 199$ kpc - - | Phat-ELVIS [10] Sigmoid-Einasto Equation (5) $\alpha_E = 0.68$ $r_{-2} = 128$ kpc $r_0 = 29.2$ kpc $r_c = 4.24$ kpc | SL17 [15] with $\epsilon_t = 10^{-2}$ $\propto \rho_{sm}$ NFW $^\star$ $r_{-2} = r_s = 19.6$ kpc $^\star$ - - | SL 17 [15] with $\epsilon_t = 1$ $\propto \rho_{sm}$ NFW $^\star$ $r_{-2} = r_s = 19.6$ kpc $^\star$ - - |
| $N_{\mathrm{calib}}$ | 300 | - | 276 $^\star$ | 276 $^\star$ |
| $N_{\mathrm{surviving}}$ | - | 90 | 114 ± 11 | 112 ± 10 |
| $c(m)$ | Moliné et al. [75] | Moliné et al. [75] | Sánchez-Conde & Prada [63] | Sánchez-Conde & Prada [63] |

$^\star$ Properties of the initial subhalos from which the surviving ones are obtained after interaction with the baryonic potential.

### 3.2.1. Model #1: DM only (as Implemented in Hütten et al., 2016)

This first configuration uses the position-dependent concentration parametrization of Moliné et al. [75]. The latter is based on the analysis of the DM-only simulations VL-II [76] and ELVIS [77], and it predicts that subhalos of a given mass are more concentrated close to the galactic center than in the outer parts of their host halos. This effect is related to tidal disruption in the DM potential of the host halo. In the outer parts, when tidal disruption in the DM potential becomes

---

[9] We still provide later the total DM profile for the semi-analytical configurations (model #3 and model #4) because it is one of the building blocks of the model.

negligible, the concentration is very close to the concentration found for field halos [63].[10] DM-only tidal effects also affect the spatial PDF of subhalos [62], and all recent DM-only simulations found it to be flatter than the DM distribution in the smooth halo. We use an Einasto profile for $\mathrm{d}\mathcal{P}/\mathrm{d}V$ with $\alpha_E = 0.68$ and $r_{-2} = 199$ kpc, following the results of the Aquarius A-1 halo [74], and we fix the number of subhalos above $10^8\,\mathrm{M}_\odot$ to $N_{\mathrm{calib}} = 300$, as an upper bound to what was found in the Aquarius simulations [74]. Subhalo outer radii and corresponding masses in this model are kept to be *cosmological* radii and masses, and subhalos are truncated where their mean density reaches 200 times the critical density of the Universe (see, e.g., [6] for details). This model is contained in the parameter space already explored in our previous study [1].

### 3.2.2. Model #2: DM + Galactic Disk Potential (Numerical, Phat-ELVIS)

In the recent Phat-ELVIS simulations, Kelley et al. [10] have accounted for the effect of the Milky Way baryon potential (including stellar and gas disk, and bulge). They found that subhalos were strongly destroyed in the inner part of the halo, leaving basically no subhalo with mass $m \lesssim 5 \times 10^6\,\mathrm{M}_\odot$ within the inner $\sim$30 kpc of the host DM halo. This is at odds with the predicted $\mathrm{d}\mathcal{P}/\mathrm{d}V$ of previous simulation sets (e.g., Aquarius [74] that was previously used as reference in [1]).

Using the subhalo catalog from the simulations provided by the authors, we compute the normalized subhalo PDF per unit volume, $\mathrm{d}\mathcal{P}/\mathrm{d}V$, at $z = 0$, averaged over the 12 Milky-Way-like halos available. The dispersion in each bin provides the error bar. The data are fitted using the following parametrization:

$$\frac{\mathrm{d}\mathcal{P}}{\mathrm{d}V}(r) = \frac{A}{1 + e^{-(r - r_0)/r_c}} \times \exp\left\{ -\frac{2}{\alpha_E}\left[ \left(\frac{r}{r_{-2}}\right)^\alpha - 1 \right] \right\}, \tag{5}$$

and results are shown in the left panel of Figure 1. The first term in Equation (5) is a Sigmoid function, centered on $r_0$ and increasing from 0 to $A$ given $r_c$, that allows us to capture the sharp decrease of the number of subhalos in the inner regions of the parent halo. The Sigmoid then transitions to an Einasto profile with characteristic scale $r_{-2}$ and index $\alpha_E$. In order to have an Einasto profile close to the DM-only case in the outer parts (see previous paragraph), we fix $\alpha_E = 0.68$, and the best-fit parameters, obtained on all subhalos in the catalog are $r_0 = 29.2$ kpc, $r_c = 4.24$, and $r_{-2} = 128$ kpc.[11] The constant $A$ is set to ensure $\mathrm{d}\mathcal{P}/\mathrm{d}V$ is a PDF.

Finally, among the 12 Milky-Way-like host halos of the Phat-ELVIS simulations, we select halos with masses similar to the NFW halo introduced in the following Section 3.2.3, in the range of $1.1 \times 10^{12} - 1.4 \times 10^{12}\,\mathrm{M}_\odot$; averaging the number of subhalos between $10^8$ and $10^{10}\,\mathrm{M}_\odot$ that have survived interaction with the baryonic potential, we fix in CLUMPY the normalization of the number of subhalos to $N_{\mathrm{calib}} = N_{\mathrm{surviving}} = 90$. In the same way as in the DM-only model # 1, we define and calculate subhalo masses based on their cosmological radii.

### 3.2.3. Models #3 and #4: DM + Disk Potential (Semi-Analytical, SL17)

Complementary to numerical approaches, semi-analytical models have also been considered by several authors, e.g., [11–15]. In this work, we use the study by Stref and Lavalle [15] (SL17 hereafter) to capture the effects of tidal stripping from the smooth galactic potential of DM and baryons and disk shocking by the baryonic disk. The model in SL17 is built on the dynamically constrained mass models from McMillan [71], where the latter used kinematic data (including maser observations, the Solar

---

[10]  The difference between using the space-dependent or field halo concentration was found to be a factor $\sim$2 larger on the brightest subhalo in [1].

[11]  The Milky-Way-like halos in the Phat-ELVIS simulations have masses ranging from $7 \times 10^{11}\,\mathrm{M}_\odot$ to $1.9 \times 10^{12}\,\mathrm{M}_\odot$, and virial radii from 235 kpc to 329 kpc respectively. The fit above was performed on the radial range common to all host halos, namely from 0 to 235 kpc. The value of the parameters remain compatible at the one-sigma level when increasing the radial range to 329 kpc, or when cutting on masses above $5 \times 10^6\,\mathrm{M}_\odot$.

velocity, terminal velocity curves, the vertical force and the mass within large radii) to determine the Milky Way's DM distribution following the parametrization of Zhao [78]

$$\langle \rho_{\text{tot}}(r) \rangle = \rho_{\text{s}} \left( \frac{r}{r_{\text{s}}} \right)^{-\gamma} \left[ 1 + \left( \frac{r}{r_{\text{s}}} \right)^{\alpha} \right]^{(\gamma - \beta)/\alpha} . \tag{6}$$

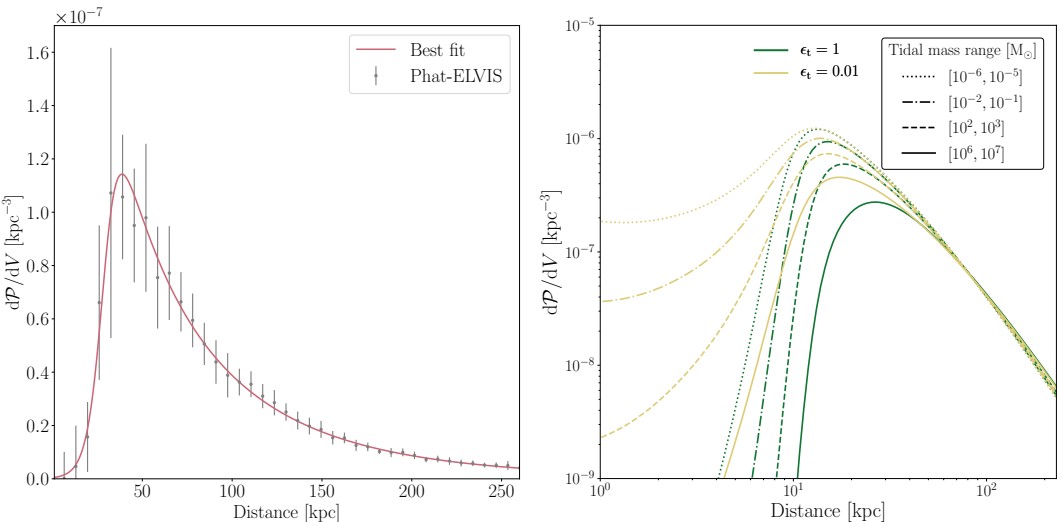

**Figure 1.** Spatial PDFs of subhalos surviving interaction with the baryonic disk potential. (**Left panel**): directly computed from the catalogs of the Phat-ELVIS simulations [10]. Dots correspond to the average over the 12 Milky-Way-like halos in the simulations, with error bars obtained from the dispersion over the 12 halos. The best-fit model (red curve) has been computed using the parametrization given by Equation (5). (**Right panel**): SL17 model for various mass ranges (line styles) and values of $\epsilon_{\text{t}}$ (colors). See Figure 2 for a comparison between all $d\mathcal{P}/dV$ models used in the analysis.

In this work, we only consider the results based on the NFW parametrization ($\alpha, \beta, \gamma = 1, 3, 1$), for which the best-fit parameters are $r_{\text{s}} = 19.6$ kpc, $\rho_{\text{s}} = 8.517 \times 10^6$ M$_\odot$ kpc$^{-3}$, resulting in $R_{200} = 231.7$ kpc and $M_{200} = 1.31 \times 10^{12}$ M$_\odot$. We checked that our conclusions are left unchanged if considering a cored profile ($\alpha, \beta, \gamma = 1, 3, 0$) instead. In SL17, the initial population of subhalos traces the above Galactic DM halo mass PDF and there is *initially* no correlation between a subhalo's mass and its position,

$$\frac{d\mathcal{P}}{dV}(r, \text{initial}) \propto \langle \rho_{\text{tot}}(r) \rangle . \tag{7}$$

The addition of tidal interactions modifies this picture because tidal effects select subhalos based on their mass and concentration to produce an 'evolved' subhalo population. This evolution manifests itself in two aspects: (i) subhalos with a given mass at a given position with too small a concentration are disrupted, while (ii) subhalos that survive are stripped from a large fraction of their mass. Mass stripping is encoded into the subhalo tidal radius $r_{\text{t}}$, which is computed as described in [15]. Please note that tidal effects in SL17 are computed assuming circular orbits for the clumps. Cosmological simulations show that subhalos are efficiently disrupted by tidal interactions. For instance, Hayashi et al. [79] find that a subhalo with tidal radius $r_{\text{t}}$ and scale radius $r_{\text{s}}$ is disrupted if $r_{\text{t}} \lesssim 0.77 \, r_{\text{s}}$. It has however been recently pointed out by van den Bosch et al. [80,81] that disruption within simulations might be largely explained due to a lack of numerical resolution, Poisson noise, or runaway instabilities. According to these authors, subhalos are far more resilient to tidal disruption than numerical simulations tend to show, implying that a subhalo could survive even if $r_{\text{t}} \ll r_{\text{s}}$ (this result is expected from theoretical grounds [82,83] and in agreement with earlier findings by

Peñarrubia et al. [84]). The SL17 model includes a free parameter $\epsilon_t$ that allows us to simply investigate both possibilities. The $\epsilon_t$ parameter is defined such that

$$\frac{r_t}{r_s} < \epsilon_t \Leftrightarrow \text{subhalo is disrupted.} \tag{8}$$

This disruption criterion can also be expressed in terms of the concentration: the subhalo is disrupted if $c < c_{\min}$ where $c_{\min}$ is referred to as the minimal concentration (which depends a priori on the subhalo's position and mass). A "cosmological-simulation-like" configuration, where subhalos are efficiently disrupted, corresponds to $\epsilon_t \sim 1$. Conversely, a model of very resilient subhalos is obtained by setting $\epsilon_t \ll 1$. In the following, we will consider two extreme values : $\epsilon_t = 0.01$ (model #3) and $\epsilon_t = 1$ (model #4). The former value implies a subhalo is disrupted when it has lost around 99.99% of its cosmological mass. Please note that the value of $r_t$ does depend on the choice of $\epsilon_t$.

The behavior of the SL17 model is illustrated in Figure 1 (right panel) where we show the spatial distributions of surviving subhalos for different mass decades. We see that the distribution of lighter objects extends to lower radii than the distribution of heavier ones. This is because, as already pointed out in [15], smaller objects are more concentrated on average and therefore more resilient to tidal disruption. Interestingly, if simulations overestimate tidal disruption as pointed out in van den Bosch et al. [80,81], i.e., $\epsilon_t = 0.01$ with our parametrization, SL17 predicts a large population of very light subhalos in the innermost regions of the Milky Way.

The number of subhalos in SL17 is calibrated onto the Via Lactea II cosmological simulations [76], by the mass fraction in resolved subhalos in these simulations. Since Via Lactea II is a DM-only simulation, the calibration is performed without the baryonic tides and setting $\epsilon_t = 1$ for consistency. This leads to $N_{\text{calib}} = 276$ for initial cosmological subhalo masses, $m_{200}$, between $10^8 \, \text{M}_\odot$ and $10^{10} \, \text{M}_\odot$ as quoted in Table 1. Adding a baryonic potential and considering tidally stripped masses leads to $\sim$110 surviving subhalos in the same mass range for both choices of $\epsilon_t$ (see also Table 1).

### 3.2.4. d$\mathcal{P}$/d$V$ Model Comparison

The four subhalo PDFs considered in this study are compared in Figure 2. The configurations based on the Phat-ELVIS and the Aquarius simulations are shown in red and blue, respectively, while the SL17 models are shown in yellow ($\epsilon_t = 0.01$) and green ($\epsilon_t = 1$). In order to make a meaningful comparison with the simulation results, we only show the SL17 prediction for large subhalo tidal masses ($m > 10^6 \, \text{M}_\odot$), comparable to the mass of the smallest objects identified in Phat-ELVIS.

The effect of a galactic disk as implemented in the Phat-ELVIS simulation is to disrupt most subhalos in the inner 30 kpc of the galactic halo, as opposed to a DM-only simulation such as Aquarius where subhalos can survive down to much lower radii. The subhalo distribution predicted by SL17, which accounts for the effect of the disk, resembles the one of Phat-ELVIS in that massive subhalos are disrupted at the center. We note that d$\mathcal{P}$/d$V$ peaks at a lower radius in SL17 compared to Phat-ELVIS. Setting $\epsilon_t = 0.01$ pushes the peak radius to an even lower value with respect to the $\epsilon_t = 1$ case, as expected since subhalos are then more resilient to disruption.

A more detailed understanding of the difference between these models is beyond the scope of this paper, since the SL17 models are semi-analytically constructed from the kinematically constrained mass model of the Milky Way from [71] taking into account the mass dependence of the subhalo spatial distribution, while the DM-only and Phat-ELVIS configurations are based on approximate fits to Milky-Way-like halos from numerical simulations. We still note a similar trend between the SL17 ($\epsilon_t = 1$) and Phat-ELVIS configurations, which may indicate that the semi-analytical method associated with the former (pending some simplifying assumptions) somewhat capture the main features of the latter (pending numerical effects likely to dominate in the central regions).

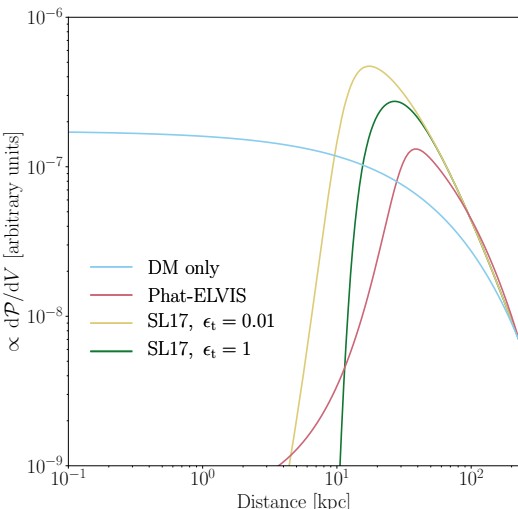

**Figure 2.** The four spatial PDFs of subhalos considered in this work: SL17 models for subhalos with $m > 10^6 \, M_\odot$ (yellow and green), PDF based on the Phat-ELVIS simulation (red) [10], and on the Aquarius simulation (blue) [74]. To highlight the behavior at low radii where tidal effects are the most relevant, the curves are shifted to match the value of $d\mathcal{P}/dV(231.7 \, \text{kpc})$ in the Phat-ELVIS configuration.

## 4. Results

Using `CLUMPY`, we generate fullsky subhalo populations and corresponding *J*- and *D*-factors according to the models from Table 1. For all configurations, the distance between the Sun and the Galactic center is set to $R_\odot = 8.21$ kpc [71]. We consider two estimations of the *J*-/*D*-factors, integrating either over the full angular extent of a subhalo or up to a radius of $\alpha_{\text{int}} = 0.5°$. Averages and PDFs are then obtained from a statistical sample of 1000 realizations of the subhalo population for each model.

### 4.1. Subhalo Source Count Distributions

Figure 3 presents the source count distribution for all models and *J*-factors $J > 2 \times 10^{16} \, \text{GeV}^2 \, \text{cm}^{-5}$ ($D > 2 \times 10^{15} \, \text{GeV} \, \text{cm}^{-2}$),[12] similar to Figure 3 of our earlier work [1]. Solid lines show *J*-/*D*-factors within $\alpha_{\text{int}} = 0.5°$ ($\Delta\Omega = 2.5 \times 10^{-4} \, \text{sr}$), dashed lines the full signal. For two configurations, we also show the variance bands of the distributions. For the first time, we also show in this work the *D*-factor distribution for decaying DM. Comparing the solid and dashed curves, it can be seen that in the case of annihilation, most of the fainter halos' emission is contained within the innermost 0.5° except for the ~100 brightest halos (left panel). This is different in the case of DM decay, for which the emission profile is much more extended (right panel). Comparing the models of this work, we reach the following conclusions for *annihilating* DM (left panel):

- Model #2 (Phat-ELVIS, red lines) predicts about a factor 5 less halos per flux decade than the Aquarius-like DM-only reference model #1. The average brightest halo (within 0.5°) is about a factor 4 fainter than expected for the DM-only case. This drastic decrease of bright objects is both attributed to the fact that the Phat-ELVIS simulations [10] find (i) overall less subhalos in Milky Way-like galactic halos ($N_{\text{calib}} = 90$ vs. $N_{\text{calib}} = 300$) and (ii), no subhalos are found close to Earth in the innermost 30 kpc of the galactic halo.
- Model #3 (SL17, yellow lines) predicts almost the same abundance of bright halos as the DM-only model #1. Model #3 starts from an initial subhalo distribution biased towards the Galactic center

---

12    We checked for convergence in order to obtain a complete ensemble of objects above these thresholds.

following the overall DM distribution, and accounts for tidal subhalo disruption and stripping afterwards according to the semi-analytical model of [15]. With $\epsilon_t = 0.01$ few subhalos are affected. In turn, the DM-only model #1 already includes a subhalo distribution anti-biased towards the Galactic center in an evolved galactic halo according to the Aquarius simulations (although the considered fitting to the Aquarius simulations [74] does not account for a mass dependence of the halo depletion).

- Model #4 (SL17, green lines) applies a much stronger condition on tidal stripping and total depletion than the model #3 configuration within the semi-analytical approach of [15]. Illustratively, we calculate a total of $1.41 \times 10^6$ initial subhalos (in the full range between $m_{min} = 10^{-6} \, M_\odot$ and $m_{max} = 1.3 \times 10^{10} \, M_\odot$) for the subhalo models #3 and #4, out of which 20,000 are completely disrupted for the model #3 ($\epsilon_t = 10^{-2}$). In contrast, 530,000 halos are disrupted for $\epsilon_t = 1$ in the model #4.[13] In result, a factor 2 less halos are present above the lower end of the displayed brightness distributions, the ratio increasing for the brightest decades. Recall that surviving halos are truncated at the same tidal radius in models #3 and #4.

For unstable, *decaying* DM, the flux is proportional to the distance-scaled DM column density (Figure 3, right panel). Contrary to the case of annihilation, we find:

- Changing the signal integration region $\Delta\Omega$ drastically impacts the collected signal, as the emission shows a much broader profile than for annihilation. This loss is most drastic for the brightest halos.
- For an integration angle of $\alpha_{int} = 0.5°$, all models are in remarkable agreement at the brightest end. For fainter flux decades and considering the signal over the full halos extent, models differ by a factor $\sim 5$ (however, with a rather large spread in the $D$-factor PDF of the brightest halo in the individual models, see the later Figure 5). This suggests that predictions for the largest subhalo flux from decaying DM should be rather model-independent.

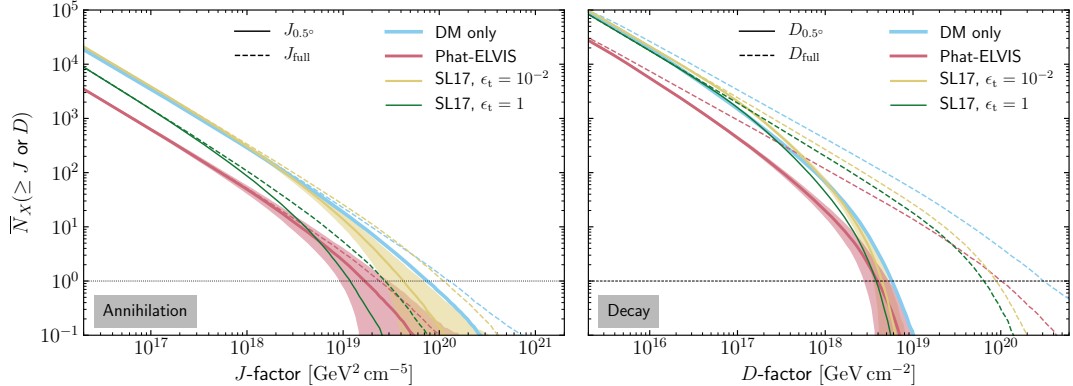

**Figure 3.** Cumulative source count distribution of galactic subhalos (full sky, averaged over 1000 simulations) for all configurations gathered in Table 1. (**Left panel**): annihilating DM. (**Right panel**): decaying DM. In both panels, the solid lines show the distribution of the $J$-factors within $\alpha_{int} = 0.5°$, whereas the dashed lines for integrating over the full halo extents up to $r_{200}$ or $r_t$.

For illustrative purposes, Figure 4 displays subhalo skymaps of a random realization of each of the four models under scrutiny. For each model and to ease comparison, the same DM subhalo sky is used in case of annihilation ($J$-factors, left) or decay ($D$-factors, right). The sky realization varies of course from one model to the other. We do not include the average and smooth DM distributions here. The maps include all subhalos with masses above $10^4 \, M_\odot$ (cosmological masses in the models

---

[13] Please note that most drawn subhalos are disrupted at masses below a tidal mass of $10^4 \, M_\odot$, the scale above which subhalos are shown in the later Figure 4.

#1 & #2, tidal masses in the models #3 & #4). For example, in the DM-only case, this corresponds to 1,214,313 halos included in the map, and for a `HEALPix` resolution of $N_{\rm side} = 1024$, `CLUMPY` requires $\sim$30 CPUh for its computation in case of annihilation ($\sim$20 CPUh in case of decay). Please note that we did not select the shown random sky realizations to reflect some particular average or extreme case. In Appendix A, we list some properties of the brightest objects in these maps, which can be compared to the average properties derived in the remainder of this section.

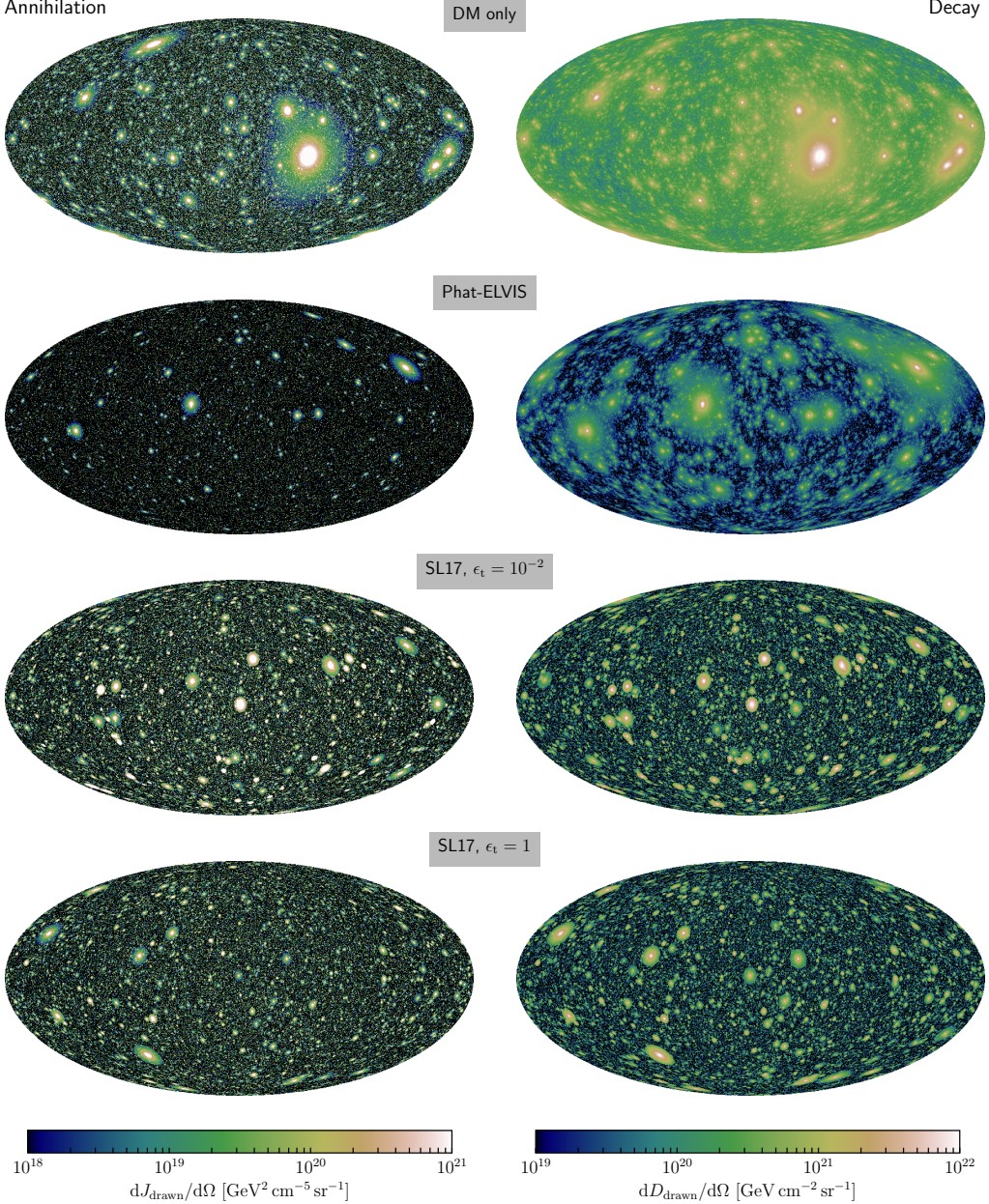

**Figure 4.** One random realization of the Galactic DM subhalo sky (all subhalos above $10^4$ M$_\odot$, ignoring the smooth contribution) in case of annihilation (**left**) or decay (**right**), derived from the models gathered in Table 1. Maps are drawn in galactic coordinates (Mollweide projection) with $(l, b) = (0, 0)$ at their centers. (**From top to bottom**): Model #1 emulating numerical DM-only simulations (1,214,313 drawn halos); model #2 emulating the Phat-ELVIS simulations [10] (364,064 drawn halos); and the semi-analytical models #3 (subhalos more resilient against tidal disruption, 549,572 surviving halos) and #4 (less subhalos surviving tidal destruction, 546,096 surviving halos) according to SL17 [15]. The displayed maps (`fits` format, 50 MB in file size) can, along with their subhalo catalogs, be provided upon request. In Appendix A, we list some properties of the brightest objects in these maps.

*4.2. Statistical Properties of the Brightest Halo*

Finally, we focus on the statistics linked to the halo with the largest *J* or *D*-factor (its properties are marked with a "$\star$" symbol in the following). For cold DM particles structuring on small scales, fully dark subhalos represent interesting targets for indirect searches. Conversely, not detecting the brightest of them can be used to set limits. We remark that focusing on the brightest halo alone for setting limits is a simplistic assumption in some circumstances. For example, there are numerous yet unidentified sources detected by the *Fermi*-LAT which could include a (i.e., the brightest) DM subhalo [16–21,85,86], so constraints set should be correspondingly weaker in this scenario.

The bottom row of Figure 5 shows the PDFs of $J^\star$ and $D^\star$, distilled from 1000 realizations of a DM subhalo sky for each model.[14] From this follows that the brightest expected signal from subhalos has in median a *J*-factor of $\widetilde{J}^\star_{0.5°} = 8.8^{+11}_{-4.0} \times 10^{19} \, \mathrm{GeV^2 \, cm^{-5}}$ ($\widetilde{J}^\star_{\mathrm{full}} = 1.6^{+2.9}_{-0.9} \times 10^{20} \, \mathrm{GeV^2 \, cm^{-5}}$) for the optimistic DM-only model. The signal is expected a factor 7 lower, at $\widetilde{J}^\star_{0.5°} = 1.3^{+0.8}_{-0.4} \times 10^{19} \, \mathrm{GeV^2 \, cm^{-5}}$ ($\widetilde{J}^\star_{\mathrm{full}} = 3.2^{+2.5}_{-1.3} \times 10^{19} \, \mathrm{GeV^2 \, cm^{-5}}$; factor 5 lower) in the case of a largely depleted inner galactic halo (model #4). For decaying DM, all models produce remarkably similar fluxes within $\alpha_{\mathrm{int}} = 0.5°$ (lower left panel of Figure 5) of $D^\star_{0.5°} \sim 4^{+3}_{-1} \times 10^{18} \, \mathrm{GeV \, cm^{-2}}$. Over the full extent of the halo, however, the width of the *D*-factor distributions is much larger, and *D*-factors between $D^\star_{\mathrm{full}} \gtrsim 10^{20} \, \mathrm{GeV \, cm^{-2}}$ up to $D^\star_{\mathrm{full}} \lesssim 10^{22} \, \mathrm{GeV \, cm^{-2}}$ are obtained.

The PDFs of the brightest subhalo's properties may also be derived. The top row of Figure 5 (left) shows that for annihilating DM, the brightest halo is found at a distance of ∼10 kpc from Earth for the models #1 (see also [1]) and #3, and also on similar Galactocentric radii (left panel of second row). For the models #2 and #4, which reflect strong tidal disruption of halos in the inner galactic region, bright halos are found at about ∼30–40 kpc distance from the Galactic center, and equivalent distances from Earth. For decaying DM, models #2 and #4 give similar predictions, while models #1 and #3 tend to predict the brightest halo at larger Galactocentric and observer distances (right panels of the top rows).

More importantly, the third row of Figure 5 sheds light on the question whether the brightest DM halo is likely to be a dark halo or a satellite galaxy. For annihilating DM, models #2 and #4 predict subhalos with masses $m^\star \gtrsim 10^8 \, \mathrm{M_\odot}$ to shine brightest in $\gamma$-rays or $\nu$; these objects are most probably associated with a (dwarf) satellite hosted in their center [88]. For the DM-only case #1 this is not anymore so obvious, as discussed in [1], as lighter objects become probable candidates to provide the highest fluxes. Finally, model #3 predicts very light and close-by halos to shine the brightest: If this model reflects the true nature of DM in our galaxy, the highest $\gamma$-rays or $\nu$ signal from Galactic DM substructure will likely arise from a dark spot in the sky. For decaying DM, the brightest Galactic DM subhalo is likely a satellite galaxy with mass $m^\star \gtrsim 10^8 \, \mathrm{M_\odot}$.

For blind searches of this brightest halo, it is finally useful to check whether there is a preferred direction in the sky to search for the brightest halo. As all our configurations are symmetrical around the direction towards the Galactic center, we present in the fourth row of Figure 5 the probability per unit area, $\mathrm{d}\mathcal{P}/\mathrm{d}\cos(\theta^\star)$, to find the brightest halo at the angular distance $\theta^\star$ from the Galactic center (GC). As found in [1], in the DM-only case #1, the brightest halo is more probably found in a direction close to the GC. In contrast, model #3 suggests to preferentially search in a ring-like region at ∼90° distance from the GC; while models #2 and #4 predict the highest probability to find the brightest subhalo towards the galactic anticenter. While these trends also apply for searches for DM decay in subhalos (right panel), they are much more pronounced for signals from annihilating DM.

---

[14] Probability distributions were derived using a kernel density estimation (KDE) with an adaptive Gaussian kernel according to [87] (except $\mathrm{d}\mathcal{P}/\mathrm{d}(\cos\theta^\star)$, which was obtained from a histogram). To handle the boundary conditions of $\mathrm{d}\mathcal{P}/\mathrm{d}m(m > m_{\mathrm{max}} = 1.3 \times 10^{10} \, \mathrm{M_\odot}) = 0$ and $\mathrm{d}\mathcal{P}/\mathrm{d}V(R > R_{200} = 231.7 \, \mathrm{kpc}) = 0$, we use the `PyQt-Fit` KDE implementation by P. Barbier de Reuille, https://pyqt-fit.readthedocs.io (not anymore maintained as of submission of the manuscript), which accounts for a renormalization algorithm at the boundary. Please note that for a precise power-law source count distribution, the PDF of $J^\star/D^\star$ follows a Fréchet distribution, see App. B of [1].

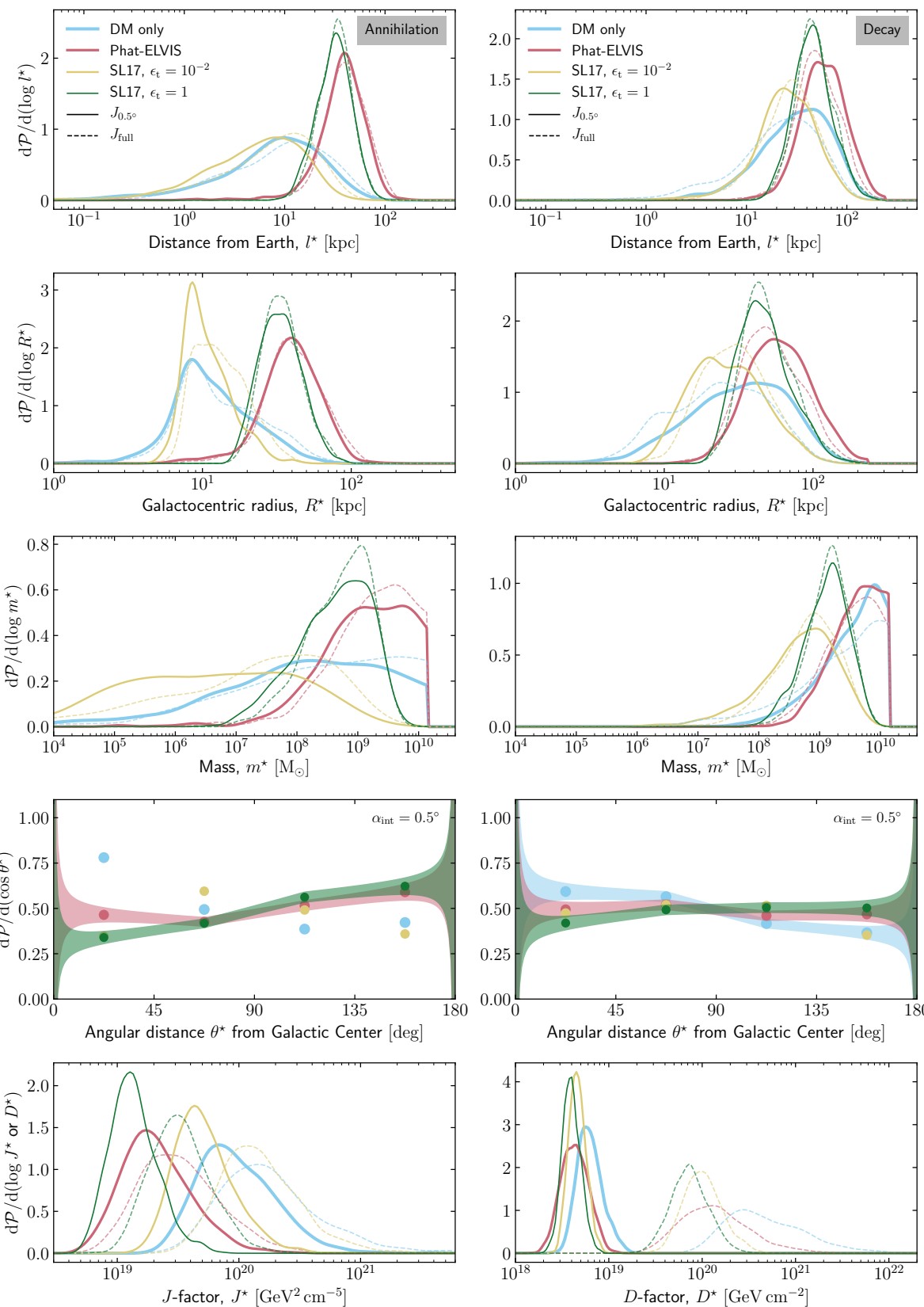

**Figure 5.** PDFs of the $\gamma$-ray (or $\nu$) brightest galactic subhalo properties for the four investigated models. (**Left panel**): annihilating DM. (**Right panel**): decaying DM. Solid lines show the statistics for only the emission from the innermost $\alpha_{\mathrm{int}} = 0.5^{\circ}$ of a subhalo, dashed lines the emission over the full extent.

## 5. Conclusions

In this work, we have studied the impact of tidal disruption of subhalos by Milky Way's baryonic potential on the properties of the $\gamma$-ray and $\nu$ signals from subhalos. This effect is mostly encoded in the spatial distribution of subhalos, and four models have been considered. The first model serves as reference and is based on 'DM-only' simulations, not including tidal disruption in the baryonic potential. Similar models based on this assumption have been applied by many authors in the past, e.g., [1,16,18,29,58]. A second model was obtained by implementing the recent results from the Phat-ELVIS numerical simulation (resolution-limited to a few $10^6\,M_\odot$) where the inner 30 kpc of the Milky Way are depleted of subhalos. The last two models relied on semi-analytical calculations applicable to the whole subhalo mass range: these calculations find a Galactocentric radius $\lesssim$10 kpc, slightly dependent on the subhalo mass, below which subhalos are stripped of their outer parts or even disrupted, depending on a disruption criterion $\epsilon_t$, taken to be to 1 or $10^{-2}$ in this study.

These models lead to significantly different brightness populations of DM annihilation and decay signals. To quantify the difference, we have simulated 1000 realizations of the subhalo population for each model. Focusing on the brightest subhalo, whose properties can be used to study DM detectability [1], we find (for an integration angle of $0.5°$) a factor 2 to 7 less signal compared to the case of subhalos in the DM-only configuration, but no significant difference for the decay signal. Our large statistical sample also allowed us to reconstruct the PDF of several properties of this brightest subhalo (mass, distance to the observer, angular distribution in the sky). In particular, the mass information indicates that in models without or little disruption in the disk potential, the brightest subhalo can be close by, and with a mass range below that of known dwarf spheroidal galaxies, i.e., a dark clump. On the other hand, in models with strong tidal disruption, the brightest subhalo is farther away, and its preferred mass is shifted to values similar to those of satellite galaxies, i.e., it could be a known dSph. The latter situation would worsen the prospects of blind galactic dark clump searches with background-dominated instruments that were discussed in [1].

In any case, our results highlight the importance of better characterizing the spatial PDF of the subhalo population, in particular by constraining further the level of tidal disruption. Although both numerical and semi-analytical approaches show the same trend in reducing the number and brightness of subhalos, there remain serious quantitative differences. We recall that it is still debated whether or not tidal disruption could be significantly amplified by numerical artifacts in simulations [80,81]. On the other hand, present-day semi-analytical methods currently rely on simplifying assumptions, some of which should be relaxed, e.g., include a more realistic distribution of orbits—see complementary studies in [89,90]. A further question is related to the possible redistribution of DM inside stripped subhalos, see, e.g., [12,91,92]. Until a clearer picture surfaces, all possibilities from weak to strong tidal effects must be equally considered for indirect DM searches. Finally, we stress again that any complete DM halo model comprising a subhalo population should be checked against kinematic constraints, which are increasingly stringent for the Milky Way in the context of the Gaia results [72,73]. This is particularly important for tidal disruption as it strongly depends on the detailed description of the baryonic components of the galaxy. Predictions or limits based on galactic halo models which do not account for these constraints should be taken with caution.

All our calculations were performed with the public code CLUMPY, and our results illustrate how this code can quickly be used to incorporate and exploit any progress made by numerical simulations and/or semi-analytical calculations. All computations and drawing of random realizations of the discussed models can be repeated at one's own account. Also, the subhalo skymaps and catalogs shown here as illustration for the various models are available upon request.

**Author Contributions:** Conceptualization, all authors; Formal analysis, M.H., M.S., C.C.; Software, all authors; Data curation, M.H.; Writing—original draft preparation, D.M. and C.C.; Writing—review & editing, all authors.

**Funding:** This work was supported by the "Investissements d'avenir, Labex ENIGMASS" and by the Max Planck society (MPG). We also acknowledge support from the ANR project GaDaMa (ANR-18-CE31-0006), the OCEVU Labex (ANR-11-LABX-0060), the CNRS IN2P3-Theory/INSU-PNHE-PNCG project "Galactic Dark Matter", and  European Union's Horizon 2020 research and innovation program under the Marie Skłodowska-Curie grant agreements N° 690575 and N° 674896.

**Acknowledgments:** We thank T. Kelley and J. S. Bullock for providing us, prior to the public release, with the Phat-ELVIS subhalo catalog. We also thank M. Doro and M. A. Sánchez-Conde for inviting us on this special issue, and for organizing the *Halo Substructure and Dark-Matter Searches* workshop held in Madrid in 2018, where discussions prompted this work. Calculations were performed at the Max Planck Computing & Data Facility at Forschungszentrum Garching. We finally thank the anonymous referees for their reviews and the valuable comments which helped to improve the quality of the manuscript.

**Conflicts of Interest:** The authors declare no conflict of interest. The founding sponsors had no role in the design of the study; in the collection, analyses, or interpretation of data; in the writing of the manuscript, or in the decision to publish the results.

## Appendix A. Properties of the Brightest Subhalos in the Example Maps

In Table A1, we list the properties of the brightest subhalos in the random realizations displayed in Figure 4. Please note that different objects may provide the largest flux for annihilation or decay and for different integration regions $\Delta\Omega$, and the largest values are marked in boldface (except for the DM-only halo, where the very same object is the brightest in all considered scenarios). Further properties of the objects (structural parameters, brightness of lower ranked subhalos, etc.) can be retrieved from the full subhalo catalogs which can be provided to the reader upon request.

**Table A1.** Properties of the brightest subhalos in the random realizations displayed in Figure 4.

| | Position in Map $(l, b)$ | Distance $l^\star$ [kpc] | Mass $m^\star$ [M$_\odot$] | $J_{0.5°}$ [GeV$^2$ cm$^{-5}$] | $J_{\rm tot}$ [GeV$^2$ cm$^{-5}$] | $D_{0.5°}$ [GeV cm$^{-2}$] | $D_{\rm tot}$ [GeV cm$^{-2}$] |
|---|---|---|---|---|---|---|---|
| DM only | $(-54°, -13°)$ | 16.3 | $4.0 \times 10^9$ | $1.8 \times 10^{20}$ | $5.5 \times 10^{20}$ | $1.1 \times 10^{19}$ | $1.8 \times 10^{21}$ |
| Phat-ELVIS | $(-141°, +31°)$ | 42.2 | $2.0 \times 10^9$ | $\mathbf{1.4 \times 10^{19}}$ | $\mathbf{2.3 \times 10^{19}}$ | $3.6 \times 10^{18}$ | $\mathbf{1.3 \times 10^{20}}$ |
| | $(+37°, +8°)$ | 69.5 | $3.8 \times 10^9$ | $1.4 \times 10^{19}$ | $2.0 \times 10^{19}$ | $\mathbf{4.1 \times 10^{18}}$ | $9.2 \times 10^{19}$ |
| SL17, $\epsilon_{\rm t} = 10^{-2}$ | $(-19°, -73°)$ | 7.8 | $3.7 \times 10^6$ | $\mathbf{3.3 \times 10^{19}}$ | $\mathbf{6.1 \times 10^{19}}$ | $2.3 \times 10^{18}$ | $7.1 \times 10^{18}$ |
| | $(-73°, +20°)$ | 54.8 | $2.3 \times 10^9$ | $1.0 \times 10^{19}$ | $2.6 \times 10^{19}$ | $\mathbf{4.1 \times 10^{18}}$ | $\mathbf{9.0 \times 10^{19}}$ |
| SL17, $\epsilon_{\rm t} = 1$ | $(+71°, -25°)$ | 30.1 | $1.3 \times 10^8$ | $\mathbf{9.0 \times 10^{18}}$ | $\mathbf{1.5 \times 10^{19}}$ | $2.3 \times 10^{18}$ | $1.6 \times 10^{19}$ |
| | $(+93°, -50°)$ | 68.5 | $3.4 \times 10^9$ | $5.2 \times 10^{18}$ | $1.4 \times 10^{19}$ | $\mathbf{3.4 \times 10^{18}}$ | $\mathbf{8.4 \times 10^{19}}$ |

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
