# Peer review of "?-ray and ? Searches for Dark-Matter Subhalos in the Milky Way with a Baryonic Potential"

_galaxies, doi:10.3390/galaxies7020060_

Reviewer 1 Report

The authors carefully assessed the impact of tidal subhalo disruption on the gamma-ray and neutrino signal predictions from DM annihilation and decay. They present four benchmark cases which encapsulate the degree of uncertainty there exists in the subhalo modelling, in particular the uncertainty due to tidal disruption of subhalos in the baryonic potential. I believe the paper is already in a very good shape for publication and I do not have any major comment. The presentation is clear and the methodology sound. 

Author Response

Dear referee,

Thank you for the very positive feedback. We are happy that you consider the draft already in a suitable state for publication. We have added some minor changes marked in boldface to the manuscript, as well as updated Figure 4 (with a previously slightly incorrect drawing of the halos).

Best Regards,
Moritz Huetten for the authors

Reviewer 2 Report

The authors have undertaken a study to compare calculate the expected gamma ray and neutrino fluxes due to the distribution and density of dark matter in the Milky-Way galaxy. This study results in a lower flux when compared with other similar studies. I find this work to be an important contribution to the ongoing study of potential detection of galactic dark matter, and I recommend this paper for publication with minor revisions.

I have a few minor comments or suggestions outlined below.

1.     In section 2.1 I would suggest further clarification on equation 1. The current format could be clarified as well as the specification of the dark matter density distribution in the text below equation 1.

2.     Page 7 section 3.2.2: Stref and Lavalle are identified as reference SF17 in section 3.2.3 after being referenced on line 185.

3.     Figure 4: Could the figure display a sample halo sky for each model that lies close to the median expected flux instead of a randomly selected model? ‘

4.     Further references should include Gianfranco Bertone,

Author Response

Dear referee,

Thank you for the positive feedback. You'll find below our answers. The corresponding changes made to the paper are highlighted in boldface ("\corrected" tag in TeX, note that at some places in the new Table A1, boldface is also for the version to publish).

Best regards
Moritz Huetten for the authors
-------------------------------

REMARK 1: In section 2.1 I would suggest further clarification on equation 1. The current format could be clarified as well as the specification of the dark matter density distribution in the text below equation 1.

ANSWER: We have clarified the condensed presentation of Eq. 1 and expanded the explanation of the contained quantities into two phrases. For the DM density, it now says "overall Galactic DM density distribution". We hope this reflects your demand for further specification.

REMARK 2: Page 7 section 3.2.2: Stref and Lavalle are identified as reference SF17 in section 3.2.3 after being referenced on line 185.

ANSWER: Thank you for pointing out this inconsistency. We have accordingly corrected that line.

REMARK 3: Figure 4: Could the figure display a sample halo sky for each model that lies close to the median expected flux instead of a randomly selected model?

ANSWER: Thank you very much for this manifest comment. In fact, we have had pondered this possibility ourselves. However, we concluded that, aside from a non-straightforward definition of "average realization" (e.g., w.r.t. the brightest halo alone or the full population), seeking for such a selection may not add much more information w.r.t. some random realization. In fact, our "real" Galaxy is also only a random realization drawn from the distributions we discuss. We therefore think the virtue of showing such maps is not to reflect an "averaged" realization, but more to illustrate a concrete situation. The statistical properties of subhalos are exactly the same whatever the map, even if the brightest object varies from a map to another one.

However, to ease the reader assessing where these shown maps are located w.r.t. the probability distributions derived & discussed later in the text, we have added a table A1 in an appendix, where we list some selected properties of the particular brightest halos in these maps. (This table also illustrates that the brightest object also differs between the considered scenario, annihilation or decay, integration angle.) Also, we now stress even clearer that extensive material can be provided to the interested reader to further work with the models at her/his own account.

REMARK 4: Further references should include Gianfranco Bertone, Wilfried Buchmüller, Laura Covi and Alejandro Ibarra (2007) and the Proceedings of Science article by Harding and Dingus for the HAWC Collaboration from 2015 among others.

ANSWER: Thank you very much for suggesting these further references. We have added them along with others in the introduction (lines 54 and 61 in the new draft).

------------------
OTHER MINOR CHANGE

Complying with your third comment, we have noted an inaccurate drawing of the skymaps in Fig.4 (particularly visible in the decay maps of the DM-only and Phat-ELVIS models). We have corrected & updated these maps in the revised manuscript.